# Nature-Based Virtual Reality Feasibility and Acceptability Pilot for Caregiver Respite

**Mohammed Owayrif Alanazi** [1,2] 🔘, **Arienne Patano** [1] 🔘, **Gary Bente** [3], **Andrew Mason** [4], **Dawn Goldstein** [1], **Sina Parsnejad** [5], **Gwen Wyatt** [1,*] 🔘 and **Rebecca Lehto** [1] 🔘

1 College of Nursing, Michigan State University, East Lansing, MI 48824, USA; alanazi4@msu.edu (M.O.A.); patanoar@msu.edu (A.P.); golds129@msu.edu (D.G.); lehtor@msu.edu (R.L.)
2 Department of Nursing, College of Applied Medical Sciences, University of Bisha, Bisha 61922, Saudi Arabia
3 College of Communication Arts & Sciences, Michigan State University, East Lansing, MI 48824, USA; gabente@msu.edu
4 College of Engineering, Michigan State University, East Lansing, MI 48824, USA; mason@msu.edu
5 Texas Instruments, 1001 Winstead Dr #305, Cary, NC 27513, USA; s.parsnejad@gmail.com
* Correspondence: gwyatt@msu.edu; Tel.: +1-(517)-432-5511

**Abstract:** Home-based informal caregivers (CGs), such as the family members and friends of cancer patients, often suffer averse emotional symptoms, such as anxiety and depression, due to the burden associated with providing care. The natural environment has been valued as a healing sanctuary for easing emotional pain, promoting calmness, relaxation, and restoration. The use of virtual reality (VR) nature experiences offers an alternative option to CGs to manage emotional symptoms and improve their quality of life. The aim of this mixed-method pilot was to evaluate the feasibility and acceptability of a nature-based VR experience for home-based CGs. Nine informal CGs participated in a 10 min nature-based VR session and completed feasibility, acceptability, and VR symptom measures in the laboratory. Semi-structured interviews with five of the CGs provided qualitative data regarding their experiences with VR. The CGs (mean age 64.78 years) were mostly female (n = 7). Our analysis showed high feasibility (15.11 ± 1.76; range 0–16) and acceptability (15.44 ± 1.33; range 0–16), as well as low VR Symptoms (1.56 ± 1.33; range 0–27). Participants primarily expressed positive perceptions regarding VR feasibility and acceptability during interviews. Our findings show promise for the use of VR nature experiences. In the next phase of the study, the intervention will be tested on home-based informal CGs of patients at end of life.

**Keywords:** nature virtual reality; caregiver support; feasibility; acceptability; mixed methods





## 1. Introduction

Approximately 43.5 million caregivers have provided unpaid care to an adult or child in the last 12 months [1]. Of these caregivers, cancer caregiving ranks high [2], with people providing care for cancer patients for 32.9 h per week on average [3]. Most often, these caregivers (CGs) are friends or family members whose responsibilities include direct patient care, medication management, communication with the healthcare team and arrangement of appointments, taking on financial responsibilities, and providing emotional support to the patient [4,5]. Home-based informal CGs often suffer averse symptoms, such as anxiety, depression, exhaustion, fatigue, and hopelessness, all of which are associated with the burden providing care [5]. Because of the physical, cognitive, and emotional demands associated with caregiving, many are starting to recognize that quality patient home-based care is dependent on CGs who have access to restorative opportunities [6,7]. Thus, it is essential to evaluate various easy-to-use and readily available supportive resources that promote the well-being of CGs and improve their quality of life. One approach to supporting CGs is through boosting their exposure to the natural environment, as it has been shown that engaging with nature is associated with better overall health [7–13]. The

use of virtual reality (VR) technology may be a promising approach in the context of caregiving, offering CGs a flexible option to manage emotional symptoms and receive much-needed respite through experiencing nature when experiencing nature in real-life is not possible [14,15].

Virtual reality technology provides an immersive three-dimensional sensory experience in visual, auditory, and spatial domains and has been used effectively in healthcare settings to treat anxiety, promote pain management through distraction, and to enhance physical rehabilitation [16–19]. While there is early evidence that the use of VR is a feasible and acceptable therapeutic approach for healthy individuals, there is limited evidence of VR being used to support stressed informal CGs. Given the limited evidence on the acceptability of providing CGs with nature-based VR experiences for therapeutic means, the purpose of our study was to evaluate the feasibility and acceptability of using a mixed-method approach in a laboratory setting. The information provided in this paper will provide the essential foundations for research as we intend to move forward with a full-scale randomized study involving home-based CGs.

The research questions addressed in the study include:

(1) Is an immersive nature-based VR experience feasible and acceptable for use with informal home-based CGs?
(2) Are adverse symptoms associated with the use of VR-delivered immersive nature experiences?
(3) Which nature scenes were most satisfactory based on user data?

## 2. Materials and Methods

### 2.1. Design

This study used a mixed-method design to test the feasibility, acceptability, and the presence/absence and severity of adverse symptoms associated with informal CGs using nature-based VR experiences. For the quantitative component of this study, self-reported surveys were used. For the qualitative component, semi-structured interviews were conducted with a subset of the quantitative sample to obtain their perceptions of the challenges of feasibility and/or acceptability benefits of incorporating nature via VR technology in their daily lives.

### 2.2. Participants

CGs were recruited via email from a large public university community. Interested CGs were encouraged to contact the study e-mail or phone number listed in the e-mail. Eligible participants were home-based informal CGs who had provided care for a partner, family member, or friend 12 months prior to entering the study. Examples of the type of informal caregiving included the sustained provision of non-paid home-based supportive care for ill relatives, elders, and/or disabled family members. CGs were excluded from the study if they had apprehension towards birds, mountains, or immersion into the deep ocean, as these were elements found in the various VR nature scenes. Nine CGs were recruited and they all completed both the nature VR experiences and the data collection process in the laboratory setting. Five of the nine CGs took part in follow-up semi-structured interviews.

### 2.3. Procedure

IRB approval from the university funding this research study was obtained prior to its initiation. Signed informed consent was obtained from all participants. Participants were instructed that they could withdraw from the study at any time during the laboratory testing and data collection.

**Introduction to VR Use and VR Session**. Upon arriving at the lab, the CG was greeted by a research assistant who proceeded to establish a rapport. The equipment was then set up for the participant. The research assistant then asked the CG to read and sign the informed consent form. After giving their informed consent, participants completed the paper-based demographic survey. Next, VR equipment was introduced, i.e., a VR headset and two hand-held controllers, which are used to navigate the immersive environment.

The participant was asked to remain comfortably seated during VR use for safety. Once participants were trained with regards to how to use the equipment, they were given time to explore the nature scenes to determine the scene that they enjoyed most. Participants were asked to select one of nine natural scenes. The VR headset contained pre-loaded scenes created to provide an immersive VR experience [20]. These scenes were designed to replicate natural environments in a visually appealing and realistic way. Each scene contains audio, such as the soothing sound of sea waves or bird songs, which aim to enhance the sense of being outdoors. Further, some scenes contained interactive parts, including wildlife such as deer and sea turtles that were present to mirror reality as close as possible. These scenes included: Black Beginning (an outer space journey for a planetary experience), Blue Deep (an underwater experience in the ocean to view sea life such as dolphins and fish), Blue Moon (an evening sky and peaceful night), Blue Ocean (a tropical beach with palm trees, surf sounds, seagulls, and turtles), Green Meadow (a spring meadow with a flowing stream and gentle wildlife), Orange Sunset (a wooded forest with tall trees and glowing red sunset skies), Red Savanna (a wide-open sunset Savanna with grazing safari animals, such as elephants), Red Fall (a fall meadow with mountains and trees with falling colorful maple leaves), and White Winter (a winter scape with snow, wildlife, and mountains in the distance) [20].

Once a nature scene was selected, the participant immersed themself in the nature scene for a full 10 min period while sat comfortably in a chair.

**Post VR Intervention**. Following the 10 min VR session, participants completed the post-test measures: feasibility, acceptability, and VR symptoms. They were thanked for their time and escorted to the lab's exit. The consent form and surveys were stored in a secure file in the research office, where each participant's file was de-identified and assigned a unique study number. An incentive gift card was sent electronically within a week.

**Follow-up Interviews**. Participants were asked via a follow-up e-mail whether they were interested in volunteering for the semi-structured interview to discuss their experiences with the VR nature experience. Examples of the interview questions include: describe your experience with the VR study; describe the components of the VR study that you found (easy or difficult) to use; describe the benefits you feel are available through using VR; and describe any unpleasant sensations that you may have experienced while using the VR technology. Those who expressed interest emailed the signed consent back to the research assistant. The research assistant coordinated and scheduled a common date and time for the interview. Interviews were conducted remotely using Zoom video-conferencing technology [21]. Each interview was conducted using a standardized set of questions and was conducted by two trained doctoral students. Questions elicited information about the VR experience and included challenges for feasibility with the use of the technology and benefits for acceptability. All interviews were audio-recorded and transcribed verbatim using Zoom's live transcription feature. Detailed descriptive field notes were also taken during the interviews for cross-checking purposes and to inform any follow-up questions. During the interview, participants were asked follow-up questions to validate their meaning and confirm the researcher's interpretation of their answers. An audit trail that included notes, dates, and time stamps also enhanced trustworthiness. A second gift card was sent electronically to the five CGs who took part in the interviews.

### 2.4. Measures

All measures were completed using paper and pencil and stored in a secure file in the laboratory.

Demographics. Demographic information was assessed by an investigator-developed survey that included questions about the year of birth, sex assigned at birth, the highest level of education, ethnic and racial background, employment status, marital status, and the presence/absence and age range (0–20 or 21 or older) of dependents living at their home.

Feasibility and Acceptability. The Feasibility of Intervention Measure (FIM) and Acceptability of Intervention Measure (AIM) surveys were used to evaluate the feasibility

and acceptability of immersive nature VR interventions [20,22]. Each of these validated measures has 4 items that are scored on a scale of 0 to 4, where 0 indicates "completely disagree" and 4 indicates "completely agree" [20]. Thus, the range for each scale extends from 0 to16, with higher scores indicating higher feasibility and acceptability. Both 4-item FIM and AIM measures have an established Cronbach's $\alpha$ of 0.85 [20]. An example item on the FIM survey included "The VR seems easy to use." In the AIM survey, an example item included "The VR is appealing to me."

Virtual Reality Symptom Experiences. This study used a modified version of the Virtual Reality Sickness Questionnaire (VRSQ) [23] to measure the symptoms one experiences while or after being in a VR environment. The 4-item scale evaluates 9 commonly experienced symptoms that have been associated with VR use (general discomfort, fatigue, eyestrain, difficulty focusing, headache, fullness of head, blurred vision, dizziness with eyes closed, and vertigo). The items are scored on a Likert scale, none (0), slight (1), moderate (2), and severe (3). The range extends from 0 to 27. The established Cronbach's $\alpha$ was 0.84 [23].

*2.5. Data Analysis*

2.5.1. Quantitative Analysis

Analyses of the survey data were carried out using Statistical Package for the Social Sciences (SPSS, v. 28.0). Descriptive statistical analysis included an evaluation of means, standard deviation, and frequencies for the three quantitative measures.

2.5.2. Qualitative Analysis

Content analysis of the qualitative data was conducted using NVivo software (v.1.6.2). Content analysis is a research technique that allows for the development of inferences by a systematic and objective evaluation of the data characteristics [24]. The methodology is well suited to the mixed-method approach used in this research study, whereby we gleaned additional acceptability and feasibility information about the VR intervention from the semi-structured interviews. This deductive process started with evaluating the interview transcripts and field notes using the research questions as a guide for key information. By reducing the interview text to content categories, codes were created for words or patterns that informed the study's research questions [24]. Two researchers verified the transcription independently prior to coding. Then, codes were developed to capture relevant information directly related to the feasibility and acceptability. Consistency of the coding process was ensured via individual discussions and then as a group to ensure content category similarities, frequency, differences across the data, and to maintain reliability in the interpretation.

Several techniques were employed to maximize the trustworthiness of the qualitative results. Credibility was ensured through investigator triangulation by having two authors collect and independently analyze the data [25,26], compare qualitative findings to quantitative findings, and document field notes. A third author reviewed the findings, and all disagreements were resolved through discussion until a unified consensus was achieved.

**3. Results**

*3.1. Quantitative Results*

Table 1 summarizes the demographic characteristics of the nine participants. The interested participants responded to the study recruitment email sent via the university email list. The sample of CGs consisted of individuals who provided care to their loved ones within the last 12 months (from the date of the present study's initiation). The CGs' mean age was 64.78 ($\pm$6.8 years), with the majority being female (n = 7). Regarding educational backgrounds, six participants completed a graduate/professional degree, and three attained 4-year college educations. All participants were white, and most were employed at least part-time. The majority were married or living with a partner.

**Table 1.** Demographic characteristics of participants (n = 9).

| Characteristics | Number |
|---|---|
| Age (M ± SD) | 64.78 ± 6.8 |
| Sex | |
|    Female | 7 |
|    Male | 2 |
| Race | |
|    White | 9 |
| Marital Status | |
|    Married/Living with Partner | 5 |
|    Divorced/Separated | 3 |
|    Widowed | 1 |
| Level of Education | |
|    Completed college | 3 |
|    Completed graduate/professional degree (post-baccalaureate degree) | 6 |
| Employment Status | |
|    Part-time | 3 |
|    Full-time | 4 |
|    Retired | 2 |

Descriptive analysis showed high feasibility (15.11 ± 1.76; range 0–16) and acceptability (15.44 ± 1.33; range 0–16), with few VR Symptoms reported (1.56 ± 1.33; range 0–27). The most reported VR symptoms were slight difficulty with focusing (n = 4), eyestrain (n = 3), and blurred vision (n = 2). There were no moderate or severe VR symptoms reported among the nine CGs. Across the nine VR nature scenes, the most selected scene was Blue Ocean (a tropical beach with palm trees, surf sounds, seagulls, and turtles) (n = 4), followed by Green Meadow (n = 3) (Table 2).

**Table 2.** Feasibility, acceptability, and VR symptom results (n = 9).

| Characteristics | | M (SD) | Range |
|---|---|---|---|
| Feasibility | | 15.11 ± 1.33 | 0–16 |
| Acceptability | | 15.44 ± 6.81 | 0–16 |
| VR Symptoms | | 1.56 ± 1.33 | 0–27 |
| General Discomfort | None | 8 | |
| | Slight | 1 | |
| Fatigue | None | 8 | |
| | Slight | 1 | |
| Difficulty focusing | None | 5 | |
| | Slight | 4 | |
| Vertigo | None | 8 | |
| | Slight | 1 | |
| Dizzy (with eyes closed) | None | 8 | |
| | Slight | 1 | |
| Blurred vision | None | 7 | |
| | Slight | 2 | |
| Fullness of head | None | 9 | |
| Headache | None | 8 | |
| | Slight | 1 | |
| Eyestrain | None | 6 | |
| | Slight | 3 | |

*3.2. Qualitative Results*

Five of the nine participants agreed to partake in a follow-up interview to better explore their perceptions about feasibility challenges and acceptability benefits associated with their VR experience. Table 3 summarizes the demographic characteristics of these participants. While the goal was to delve deeper into the feasibility and acceptability of the VR experience, interview questions were worded to engender this information without prompting participants directly to these concepts. For example, for the feasibility items, questions were asked about their experience with technology and whether they had any challenges while using the VR technology. For acceptability, questions regarding satisfaction level and potential benefits were asked. No interview surpassed 25 min.

**Table 3.** Demographic characteristics of participants in qualitative interview (n = 5).

| Characteristics | Number |
|---|---|
| Age (M $\pm$ SD) | 66.2 $\pm$ 9 |
| Gender | |
|     Female | 3 |
|     Male | 2 |
| Race | |
|     White | 5 |
| Marital Status | |
|     Married/Living with Partner | 3 |
|     Divorced/Separated | 2 |
| Level of Education | |
|     Completed college | 1 |
|     Completed post-baccalaureate degree | 4 |
| Employment Status | |
|     Part-time | 1 |
|     Full-time | 2 |
|     Retired | 2 |

Three major content categories were identified in the data that were descriptive of participants' VR study perceptions: (1) prior orientation with VR (technology use); (2) ease of use; and (3) affective experiences associated with the VR nature scenes. Two content categories addressed feasibility, and the third addressed acceptability.

3.2.1. Feasibility

**Content Category 1: Prior Orientation with VR and Technology.** Participants spoke extensively about their VR experience by connecting it to prior encounters with VR or similar technologies. Although all of the participants interviewed reported that they had no prior VR experience, 40% had observed someone else using VR and 60% owned and operated technical equipment with similarities to VR (i.e., Nintendo Wii, interactive computer, etc.). These prior experiences familiarized and oriented them to move forward with trying the VR study, while also increasing their comfort level with a unique experience. These findings contributed to the feasibility of VR with CGs.

For example, one participant said:

"My only prior experience with [the VR] had been secondhand, sort of observational. [Relative] has grandchildren that have VR headsets, and I've watched them with some amusement . . . so I've kind of seen how it works externally, and that seemed to prepare me to do it myself".

**Content Category 2: Ease of use.** The positive feedback received from participants regarding the ease of use of the VR headset and controllers contributed to the overall high feasibility of the VR nature experience as a respite tool for caregivers. These comments indicated that participants found the setup easy to understand, the application straightfor-

ward for a novice, and the lowered complexity to be helpful, which helped to increase the overall feasibility of the intervention.

For example, one participant said:

"Very well organized. [The research team] took me through. They got me set up with the [headset] and let me go through the different environments that were available, and I was able to choose the one that I liked best. So, I chose the ocean, and it was very easy to use. I just sat there and looked around [...] Once [the headset] was on, it was very user friendly".

The feedback from participants offers valuable insights into the feasibility of using VR in research with CGs. The fact that participants had seen the VR equipment before positively contributed to feasibility and the ease of use.

### 3.2.2. Acceptability

**Content Category 3: Affective Experience Associated with VR Scenes.** In general, the immersive VR nature scenes elicited positive emotions. For example, the nature scenes promoted a positive attitude, with the majority of participants describing the experience as "pleasant," "pleasurable," and "peaceful." All five participants reported that the experience had a relaxing or calming effect. Most addressed the plausibility of incorporating VR as a complement to their daily routine, especially as a stress reduction tool or as a coping strategy technique. All participants were welcoming and approving of the VR, indicating the acceptability of the intervention.

Several participants demonstrated the acceptability of the intervention through discussions of meaningful experiences that are associated with VR nature scenes. One participant shared that the VR beach scene was reminiscent of when she celebrated her anniversary and retirement by going on a family trip to Hawaii. The participant remarked:

"[The VR] created a positive attitude [ . . . ] it took me back to a happy time of being at the beach, and I mean it was, it was pleasurable".

Similarly, another participant said:

"I would imagine that [the VR nature intervention] could be a nice relaxation technique or coping technique".

These types of comments during the interviews confirmed the acceptability of data obtained through the quantitative data analysis. Participants seemingly related to the VR experience, which made it a very acceptable experience. There was no mention of VR symptoms during the interviews.

Overall, in response to our research questions, the quantitative data from the study showed high levels of feasibility and acceptability with regards to using VR nature experiences as a respite tool for caregivers, with a low number of VR symptoms being reported. These findings were further supported by the qualitative data from interviews, which elucidated participants' in-depth perceptions of the feasibility and acceptability of the intervention. The consistency between the quantitative and qualitative findings indicated a strong level of support for the potential use of VR technology in future interventions involving home-based informal CGs of various types of patients, including cancer or hospice patients.

## 4. Discussion

Our study findings provide promising evidence that suggests that nature-based VR scenes are feasible and acceptable options for providing respite to CGs, with the qualitative data revealing notable insights. While the quantitative data indicated few VR symptoms, none were mentioned at all during the semi-structured interviews.

The qualitative feedback from participants further strengthens the feasibility and acceptability of immersive nature-based VR interventions. Most participants were able to use the VR headsets successfully and reported that the technology-mediated nature experience was easy to manipulate and navigate. The study team learned that the VR training had to be individualized since participants learn in different ways. Some may do

fine with only verbal direction, while others do not grasp a proper understanding until they are fully experiencing the VR. Assessing the participants' level of comfort with the VR equipment is important prior to independent use by participants.

Building upon the existing literature, this work aligns with previous studies that have demonstrated the potential benefits of VR nature experiences [9]. A study examining the use of VR among middle-aged and elderly adults (n = 34) found that, while experiencing VR natural settings, participants expressed positive emotions such as pleasure, calmness, delight, relaxation, and comfort [14]. They also reported feeling a sense of freedom and being refreshed by the VR nature experience. In a study by Flynn et al. (2022) [27], an interactive VR experience involving grasping, picking up and repositioning items was provided to both people with dementia and their CGs. This experience facilitated an engaging and interactive activity and improved the social health and well-being of both the patients and CGs. These are especially important findings for a novel VR intervention since effective strategies could bridge the gap in promoting the health and well-being of many types of CGs and perhaps more so for cancer caregivers, who are often limited to indoor settings [5].

In line with the current study's findings, Yu et al. (2020) found that symptoms related to using a VR headset were scarce among participants [14]. The current study also found that only a few participants reported mild VR symptoms such as difficulty focusing, eyestrain, and blurred vision. Thus, the fact that reports of VR symptoms were scarce suggests that VR nature experiences have the potential to provide a safe and low-risk intervention for CGs, especially those who may not have access to nature outdoors.

The qualitative findings of this study strongly support the strong feasibility and acceptability of VR nature experiences. When reviewing the qualitative findings in terms if feasibility, participants described VR as easy to use. However, all participants who engaged in the interview portion of the study reported having 'prior orientation' via observing others using VR or due to the fact that they consider themselves to be technology literate, i.e., familiar with other similar technologies prior to participating in the study. Participants' prior experiences informed them of what to expect with respect to the VR equipment (i.e., headset, hand controllers) and subsequently facilitated the easy operation of the VR equipment during the study. These past experiences contributed to feasibility.

Acceptability was reported in the qualitative interviews; all participants found using VR relaxing and enjoyable and reported they would be interested in using VR headsets as part of their caregiving routine. Further, the use of VR was reported to be an effective distraction from their current caregiving situation and something which allowed them to still be nearby enough to respond to patient needs. Many participants described their VR experience as a welcome interlude or a form of escape. This positive diversion allowed them to recall tranquil nature-based memories that were significant to them. Among all CGs, cancer patient CGs are often in need of respite [4]. Virtual reality nature immersion could be complementary to the daily routines of CGs, providing a source of relief by evoking memories of a past pleasant nature-based experience or satisfying a desire for travel.

The use of VR nature experiences may offer an adaptable and accessible alternative to traditional respite options, such as travel or outdoor activities. This will be especially true once VR becomes more available through library loans or clinics, which would mean the CGs could use VR while the patient receives chemotherapy or at home. Interventions that support the well-being of CGs have the potential to improve patient care quality. Furthermore, evidence has shown that the benefits of indoor nature experiences may offer long-term advantages to one's cognitive ability, physical and psychological health, and overall life satisfaction [8].

## 5. Limitations

As with all studies, this study has limitations. Firstly, this pilot study had a very small number of participants. This meant that only trends could be identified. A larger sample would have laid a more robust groundwork for the next stage of testing. For example,

during the semi-structured interviews, one participant commented that the study had room for improvement with regards to the setup and instructions providing for VR use. In this regard, he suggested that staff should consider the intensity of the training session before the 10 min VR viewing.

This comment highlights the need for guidance or instructions to be provided by research staff to participants when setting up the VR equipment in order to ensure that individuals are effectively trained on how to use the VR headset and hand controls competently. Some individuals may also require more time to understand how to navigate and operate the VR controls. This comment helped the research team better understand that each participant has a different learning style regarding psycho-motor skills and that these individualized differences must be taken into account.

There are a few limitations related to VR technology worth mentioning. A few participants reported mild symptoms associated with using the VR equipment, such as difficulty focusing, blurred vision, or eyestrain. However, these symptoms were generally mild and resolved immediately after the intervention. Another limitation associated with using VR technology is its relative novelty, the need for access to headsets, and higher costs, which may affect the availability and accessibility of such technology for use among CGs.

## 6. Conclusions

Overall, the findings from this mixed-method pilot study suggest that nature-based experiences delivered through VR could effectively emulate the sense of being outdoors, potentially providing a source of respite for CGs. This study has shown that using nature-based VR experiences for therapeutic means is both highly feasible and acceptable among home-based CGs. The results indicate that VR headsets are easy to use and are associated with few user symptoms. Additionally, in the aforementioned interviews, participants suggested that they found VR easy to use (feasibility) and that they had a positive experience overall (acceptability). These initial findings will lay the groundwork for us to move forward to the next phase of our overall study, in which we will aim to consider diverse groups of CGs and account for the social determinants of health, such as age, sex, and ethnicity. Our future research may also focus on the type of patients receiving care, such as those with advanced cancer and those in hospices. Healthcare professionals may recommend the use of VR technology for both CGs and patients. Integrating this technology into clinical practice would offer more therapeutic options that may help reduce the burden on CGs and thus potentially enhance patient care quality. To make VR technology more accessible for CGs and patients, VR headsets could be provided through library loans or clinic initiatives. While our findings are encouraging, longitudinal studies with larger and more diverse samples are needed. Our future research will also focus on the most effective ways to incorporate VR into clinical practice, specifically for CGs.

**Author Contributions:** Conceptualization, R.L. and G.W.; Methodology, R.L.; Software, M.O.A. and S.P.; Validation, R.L.; Formal Analysis, M.O.A., G.W., D.G. and A.P.; Resources, G.B. and A.M.; Data Curation, M.O.A., G.W. and A.P.; Writing—Original Draft Preparation, M.O.A., G.W. and A.P.; Writing—Review and Editing, M.O.A., G.W., A.P., D.G. and R.L.; Visualization, M.O.A. and A.P.; Supervision, R.L. and G.W.; Project Administration, M.O.A. and G.W.; Funding Acquisition, R.L. and G.W. All authors have read and agreed to the published version of the manuscript.

**Funding:** This research was funded by the Michigan State University Trifecta grant number: GA013811.

**Institutional Review Board Statement:** The study was conducted in accordance with the guidelines of the Declaration of Helsinki and approved by the Institutional Review Board of Michigan State University (STUDY00005799, 26 September 2022).

**Informed Consent Statement:** Written informed consent was obtained from all subjects involved in the study.

**Data Availability Statement:** The data presented in this study are available upon request from the corresponding author. The data are not publicly available due to privacy.

**Acknowledgments:** We would like to acknowledge the valuable contributions of Grace Caldwell from the College of Nursing, Andrew Klerk from the College of Natural Sciences, Department of Integrative Biology, Nolan Jahn from Communication, and Raed Hailat from the department of Epidemiology and Biostatistics at Michigan State University.

**Conflicts of Interest:** The authors declare no conflicts of Interest.

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
