# Peer review of "Nature-Based Virtual Reality Feasibility and Acceptability Pilot for Caregiver Respite"

_curroncol, doi:10.3390/curroncol30070448_

Round 1

Reviewer 1 Report

Article explores one of the most important negative effects in caregiving, especially in those care paths which involves cancer patients and home care services: the burden of care. Apart from common support strategies, such as psychological support, new technologies may provide new methods which have to be tested. The use of virtual reality to support caregivers is a cutting-edge research topic, but the methods by which to investigate it must be rigorous and the research design should be well designed. Below are my comments on the article.

-          The sample on the quantitative side is critically small (9 informal caregivers). Since any quantitative inference or generalization may be methodologically biased, I would rather suggest to focus on qualitative results.For the same reason, I recommend not using percentages in tables and comments.

-          Since it could be argued that simply being interested in a caregiver's health is in itself a way to support caregivers itself, to correctly test hypothesis “the nature-based VR is a way to provide respite to informal CGs” it would require to set a study to compare positive effects on treated and not treated populations.

-          On qualitative side, since only half of the participants agreed to conduct interviews, results could be affected by an uncontrolled selection bias: those CGs with positive experiences agreed to participate, then CGs with negative ones self-excluded.

-          All comments reported in qualitative focus are positive and all CGs had, directly or indirectly, previous experience on the VR or similar technologies. Here there is another possible sort of framing biases: agreeing on participate in VR experience in the pilot carried out by authors and reporting positive experience may be correlated. It would be interesting deepen the qualitative analysis on critical aspects of the experiences and contextually suggest possible solution to ameliorate the VR experience.

-          - One of the aspects that can promote the use of VR experience to help CGs may concern the duration of emotional support provided by the VR experience. Does the author have any idea about the duration of the alleged positive effects?

Reviewer 2 Report

The abstract of the study appears to be well-structured. It provides a clear overview of the pilot study's objective, methods, and results on the feasibility and acceptability of nature-based virtual reality (VR) experiences for home-based informal caregivers (CGs) of cancer patients. The abstract effectively highlights the potential benefits of using VR nature experiences as an alternative option for managing emotional symptoms and improving the quality of life for CGs.

Few areas where the abstract could be improved:

Clarify the specific emotional symptoms and burdens associated with care provision that the CGs experience.

Provide a bit more context regarding the sample of CGs

The results section summarizes the quantitative findings related to feasibility, acceptability, and VR symptoms.

Overall, the abstract provides a good overview of the study, but incorporating the suggested improvements will enhance the clarity and completeness of the information presented.

The introduction provides sufficient background information and includes relevant references. It highlights the prevalence and significance of caregiving, specifically in the context of cancer caregiving, and emphasizes the physical, cognitive, and emotional burden home-based informal caregivers face.

To improve: The introduction briefly mentions the potential of virtual reality (VR) technologies to support individuals in managing emotional concerns but does not provide a clear transition to the research questions addressed in the study. Consider adding a sentence or two that directly connects the use of VR technologies to the context of caregiving and the potential benefits for home-based informal caregivers.

The research design appears to be appropriate for the study's objectives. The methods are adequately described, providing a clear overview of the study's design, participants, and procedures. The procedure section outlines the steps followed in the study, including obtaining IRB approval and obtaining informed consent from participants.

The methods section provides sufficient details to understand the study's design, participants, procedures, and measures. However, it would be helpful to include more information on the specific questions asked during the semi-structured interviews and the criteria for coding and analyzing the qualitative data to enhance further the study's transparency and replicability/transferability (Lincoln and Guba criteria).

Overall, the results section presents the quantitative and qualitative findings separately. The demographic characteristics of the participants are summarized in tables, and the feasibility, acceptability, and VR symptoms are presented with mean values and percentages. The qualitative results are described in content categories with supporting participant quotes. The presentation of results seems clear and comprehensive, providing a detailed understanding of the findings.

Discussion:

Provide more details about the specific nature-based scenes used in the VR experiences. Did participants have a choice in selecting the scenes? How were the scenes designed to evoke positive emotions and a sense of being outdoors?

Discuss the potential long-term effects of VR nature experiences on the well-being and mental health of CGs. Are there any indications that the benefits may extend beyond the immediate respite during the VR sessions?

Address any potential concerns or limitations related to the VR technology itself. For example, discuss any potential discomfort or limitations reported by participants that may impact the widespread adoption of VR nature experiences.

Conclusion:

Emphasize the importance of integrating VR nature experiences into clinical practice for CGs and highlight the potential impact on patient care quality. Discuss how healthcare professionals can incorporate VR interventions into comprehensive caregiving strategies.

Address the need for further research to explore the effectiveness of VR nature experiences in different contexts, such as advanced cancer and hospice care. This can help determine if the benefits observed in this study can be replicated in diverse CG populations.

Highlight the potential policy implications of using VR technology for CG support. Discuss the feasibility of making VR headsets more accessible through initiatives like library loans or clinics, which can enhance the availability of VR experiences for CGs who lack access to natural environments.

Round 2

Reviewer 1 Report

Authors provided answers to all my comments and modified manuscript shows some improvements. Scientific soundness honestly still not great, but editor could find it acceptable as a pilot study.

I am not native English speaker, but I suggest text check. For instance, sentence: “The qualitative findings strongly support the strong feasibility and acceptability of VR nature experiences” (p. 8, lines 316-17) could be ameliorate.